# The Estrogen–Immune Interface in Endometriosis

**DOI:** 10.3390/cells14010058

**Published:** 2025-01-06

**Authors:** Emily Greygoose, Pat Metharom, Hakan Kula, Timur K. Seckin, Tamer A. Seckin, Ayse Ayhan, Yu Yu

**Affiliations:** 1Curtin Health Innovation Research Institute (CHIRI), Faculty of Health Sciences, Curtin University, Perth, WA 6102, Australia; 2Curtin Medical School, Faculty of Health Sciences, Curtin University, Perth, WA 6102, Australia; 3Department of Obstetrics and Gynecology, Faculty of Medicine, Dokuz Eylul University, Izmir 35340, Turkey; 4Burnett School of Medicine, Texas Christian University, Fort Worth, TX 76104, USA; timurseckin@me.com; 5Department of Gynecology, Lenox Hill Hospital, and Hofstra University, New York, NY 10075, USA; 6Department of Tumor Pathology, Hamamatsu University School of Medicine, Hamamatsu 431-3192, Japan; 7Department of Gynecology and Obstetrics, Johns Hopkins University School of Medicine, Baltimore, MD 21205, USA; 8Discipline of Obstetrics and Gynaecology, Medical School, University of Western Australia, Crawley, WA 6009, Australia

**Keywords:** endometriosis, estrogen, immune cells

## Abstract

Endometriosis is a gynecologic condition characterized by the growth of endometrium-like stroma and glandular elements outside of the uterine cavity. The involvement of hormonal dysregulation, specifically estrogen, is well established in the initiation, progression, and maintenance of the condition. Evidence also highlights the association between endometriosis and altered immune states. The human endometrium is a highly dynamic tissue that undergoes frequent remodeling in response to hormonal regulation during the menstrual cycle. Similarly, endometriosis shares this propensity, compounded by unclear pathogenic mechanisms, presenting unique challenges in defining its etiology and pathology. Here, we provide a lens to understand the interplay between estrogen and innate and adaptive immune systems throughout the menstrual cycle in the pathogenesis of endometriosis. Estrogen is closely linked to many altered inflammatory and immunomodulatory states, affecting both tissue-resident and circulatory immune cells. This review summarizes estrogenic interactions with specific myeloid and lymphoid cells, highlighting their implications in the progression of endometriosis.

## 1. Introduction

The endometrium is a highly complex organ characterized by significant heterogeneity amongst individuals and across various physiological functions. Understanding its dynamic changes throughout the menstrual cycle and across various physiological functions in healthy conditions is essential, particularly in the context of intricate and complex illnesses like endometriosis.

Despite advancements in single-cell and spatial multiomics technologies, which have enhanced our insights into endometrial cell interactions, challenges persist. Issues such as cell isolation, low sensitivity to rare ribonucleic acid (RNA) populations embedded within extracellular matrices or tightly associated with neighboring cells, and non-uniform annotation continue to hinder our ability to assess transcript distribution and RNA processing roles. Integrating multiomics data, particularly combining transcriptomics and proteomics, may offer significant promise for identifying novel biomarkers and elucidating the pathological mechanisms of endometriosis. By analyzing mRNA expression (from transcriptomics) and protein abundance (from proteomics) in endometriosis tissues, differentially expressed genes and proteins can be identified, advancing diagnostic markers and insights into disease progression. This approach also enables pathway analysis, revealing disrupted molecular pathways and protein–protein interaction networks, which are crucial for understanding the disease pathophysiology and identifying potential therapeutic targets. Moreover, multiomics can help address disease heterogeneity by uncovering distinct RNA and protein expression patterns across endometriosis subtypes. However, several challenges must be addressed to fully realize the potential of multiomics in endometriosis research. Data complexity, tissue availability, and the inherent heterogeneity of the disease complicate the interpretation of results. In particular, the integration of diverse omics data requires robust technical standardization. Developing standardized protocols for sample collection, processing, and data analysis is essential to ensure reproducibility and enable cross-study comparisons. These efforts are critical for advancing both the diagnostic and therapeutic potential of multiomics in endometriosis. Still, without a comprehensive grasp of normal physiological states, elucidating disease mechanisms, especially in endometriosis, becomes exceedingly difficult.

This gap in knowledge is likely a significant factor contributing to our limited understanding of endometriosis pathophysiology, which is crucial for developing effective treatments and identifying specific pathological alterations. Therefore, there is an urgent need for multi-institutional, multidisciplinary studies to explore spatial and temporal expression patterns that govern differentiation, homeostasis, and pathological states, ultimately advancing our understanding and treatment of this prevalent condition.

Recognition of the molecular mechanisms underlying endometriosis is vital, particularly the factors that trigger lesion formation. Investigating the genes and pathways involved in the attachment, growth, and persistence of endometrial tissue at ectopic sites, alongside those in the eutopic endometrium of affected individuals, is a critical focus area, especially regarding immune system dysfunction. While documented abnormalities in immune cells and cytokine activity are known to contribute to the disease, it remains unclear whether these changes are causative or resultant.

This literature review aims to enhance our understanding of hormonal–immune interactions in endometriosis by critically analyzing and summarizing the existing research. We have used the following search terms in PubMed database: endometriosis AND immune cells; estrogen OR oestrogen AND Uterus; endometriosis NOT (estrogen OR oestrogen); (endometriosis OR estrogen OR oestrogen) AND immune cells; (estrogen OR oestrogen OR estrogen receptor OR oestrogen receptor) AND (endometriosis OR immune cells); endometriosis lesions AND immune cells; endometriosis AND immune cells NOT (estrogen OR oestrogen OR estrogen receptor OR oestrogen receptor); (estrog* OR oestro*) AND immune cells; (“immune cells” NEAR/estrogen) OR (“immune cells” NEAR/oestrogen). The searches also included additional terms (in the above same formatting): ‘autoimmunity’, ‘monocytes’, ‘leukocytes’, ‘uterus’, ‘menstrual cycle’, ‘ovarian cycle’, ‘endometrial’, and ‘female immunity’. Additional specific searches include for endometrium OR endometriosis AND specific immune cells (Neutrophil, Basophil, T cells, etc.). This review presents relevant theories, concepts, and findings to identify research gaps and to provide a foundation for future studies.

We will first explore the structure, function, and physiological changes in the normal endometrium throughout the ovulatory menstrual cycle, followed by current knowledge gaps related to immune–hormonal interactions in both healthy and endometriosis-affected tissues, including the eutopic endometrium. Endometriotic lesions comprise endometrial-like tissue fragments and immune cells that likely evade destruction during retrograde menstruation. The survival and proliferation of ectopic endometrial cells is sustained by impaired immune surveillance and abnormal cytokine expression; this facilitates invasion into the mesothelial lining whilst promoting growth and neovascularization. Macrophages play a pivotal role, with their normal functions altered during both the initiation and persistence of the disease.

After providing an overview of estrogens and their isoforms, we will conclude by discussing hormone interactions with specific myeloid and lymphoid cells, highlighting the recent literature on hormone–immune interactions in endometriosis.

## 2. Human Endometrium, Physiology, and Anatomy

The human endometrium features the following two distinct yet interdependent layers: the luminal “functionalis” and the foundational “basalis”. The functionalis, a dynamic layer lining the uterine cavity, evolves throughout the menstrual cycle, comprising epithelial and stromal components, intricate glands, resilient stromal cells, spiral arteries, and various immune cells that create a nurturing environment. During pregnancy, the functionalis transforms into a supportive bed for the embryo, establishing a vital fetal–maternal connection. Below this active surface lies the basalis layer, home to progenitor stem cells that ensures self-renewal and which enables monthly regeneration. Together, these layers showcase the endometrium’s remarkable adaptability and regenerative capacity, inviting the deeper exploration of reproductive health. Since the 1950s, it has been understood that endometrial morphology corresponds directly to circulating changes in estrogen (E2) and progesterone (P4) across three distinct phases in a 28-day cycle—a phenomenon that recurs approximately 451 times during a woman’s reproductive years (assuming a typical reproductive duration of about 34.7 years) [1,2]. Each phase—menstruation, proliferation, and secretory—unfolds with rhythmic precision, driven primarily by E2 and 4, along with regulators like the hypothalamic–pituitary–ovarian (HPO) axis. In the proliferative phase, rising E2 levels stimulate rapid growth by increased mitotic activity in epithelial and stromal components and angiogenesis, creating delicate-walled blood vessels. Following ovulation, the secretory phase is marked by increased P4, thickening the endometrium through stromal decidualization, enhancing the epithelial secretory function, resulting in a convoluted structure, and blood vessel remodeling into prominent thick-walled spiral arteries that is vital for nourishing the enriched endometrium, priming it for potential implantation [3,4]. The symphony of postovulatory hormonal changes is so unique that a postovulatory day can be accurately, precisely, and reliably assessed histopathologically, based on stromal and epithelial alterations.

The adaptive response is crucial for supplying blood to the thickened endometrium, facilitating fertilized egg implantation. Conversely, if fertilization does not occur, declining P4 levels trigger tissue ischemia and necrosis, leading to menstruation and restarting the cycle. This dynamic hormonal interplay not only prepares the body for potential new life but also highlights the complex biological processes governing fertility, weaving a compelling narrative of resilience and renewal.

## 3. Immune Cells of Human Endometrium in Physiological Conditions

Endocrine and immune factors are crucial in regulating endometrial functions throughout the menstrual cycle, but the underlying dynamic cellular mechanisms remain poorly understood. At the heart of harmonious changes is a complex interplay between the endocrine and immune systems. The immune environment of the endometrium is significantly influenced by E2 and P4, reflecting the complex interplay; in fact, it is also regulated by the HPO axis and its associated immune microenvironment. Throughout the cycle, E2 promotes endometrial proliferation and selectively recruits specific immune cell populations, while P4 provides crucial immunosuppression during the preimplantation phase, creating a conducive environment for potential pregnancy, even though many menstrual cycles do not culminate in successful implantation. Moreover, E2 promotes an additional immune tolerance during pregnancy, ensuring optimal conditions for fetal development. This intricate balance between hormonal action and immune regulation underscores the remarkable adaptability of the endometrium, setting the stage for successful reproduction [5].

The composition of immune cells in the endometrium, particularly the tissue-resident ones, along with the balance and interactions of pro- and anti-inflammatory cytokines essential for proper function, is subject to fluctuations throughout the menstrual cycle (Figure 1). During the proliferative phase, counts of natural killer (NK) cells and certain T cells gradually increase, in addition to enhanced angiogenesis and tissue regeneration. However, lymphocyte numbers appear to be fewer, leading to the interpretation of a less active immune environment compared to the secretory phase. Immune cell counts rise significantly during the late secretory phase and menstruation, contributing to tissue remodeling, menstrual breakdown, and repair [6].

Endometrial immune cells, marked by the CD45 surface protein found on most hematopoietic cells, make up 10–20% of endometrial cells, peaking during the secretory phase. T cells, identified by CD3 positivity, dominate in the proliferative phase, but by the mid-to-late secretory phase, uterine NK (uNK) cells identified by CD56 positivity become predominant, comprising 70–80% of endometrial leukocytes [8]. T cells are essential for maintaining the mucosal barrier and local immune defense [9]. Key endometrial leukocytes, other than T and uNK cells, include macrophages (Mφs, CD68 positivity), along with dendritic cells (DCs, ITGAX positivity), B cells (CD20 positivity), mast cells (MS4A2 positivity), and neutrophils, which are absent in the proliferative and early secretory phases but surge before menstruation, influenced by steroid hormones [10,11].

Research underscores the critical role of immune responses during menstruation, where Mφs, NK cells, and cytotoxic T cells orchestrate tissue destruction and regeneration. These cells are carefully regulated to target menstrual remnants while preserving the integrity of regenerating endometrial tissue, thereby maintaining homeostasis. Uterine macrophages (uMφs) and uNK cells engage extensively with stromal, endothelial, and perivascular cells to support endometrial regeneration. uMφs play a pivotal role in clearing debris, sustaining tissue homeostasis, and exhibiting phenotypic plasticity to adapt to the local microenvironment. The traditional M1 pro-inflammatory and M2 anti-inflammatory classification of Mφ simplifies their diverse roles. In pregnancy, the balance of Mφ polarization at the maternal–fetal interface is crucial for maintaining a healthy pregnancy. uNK cells, briefly divided into uNK1-3, provide localized immune protection and, distinct from their peripheral counterparts, are classified as endometrial or decidual based on their location and pregnancy status, and contribute to uterine remodeling and immune tolerance so that the abnormalities in their function or numbers could lead to pregnancy complications. Endometrial stromal cells express matrix metalloproteinases (MMPs) during the menstrual phase, along with integrins and key cytokines like CCL5 (RANTES, Regulated on Activation, Normal T cell Expressed and Secreted, a pro-inflammatory chemokine that recruits leukocytes to inflammation, also chemotactic for eosinophils, basophils, monocytes, T cells, NK cells, DCs, and mastocytes) and Retinoic Acid Receptor Responder Protein 2 (RARRES2, a cell surface receptor that stimulates Mφ and DC chemotaxis while inhibiting growth) to facilitate the recruitment of uMφs to the tissue. Both uMφ1 and uMφ2 upregulate Platelet-Derived Growth Factor Subunit B (PDGFB), which is essential for wound healing, as it binds to PDGFRB on stromal cells post-menstruation [12,13].

uMφ1 upregulates tumor necrosis factor-α (TNF-α), which activates ER signaling in endometrial epithelial cells to regulate the secretion of inflammatory substances—a process inhibited by P4. It also increases Epiregulin (EREG), a member of the EGF family involved in inflammation, wound healing, oocyte maturation, and cell proliferation. In contrast, uMφ2 elevates insulin-like growth factor 1 (IGF1). Together, these factors promote the proliferation and survival of endometrial stromal cells. Additionally, immunoregulatory genes such as IL10, Galectin-9 (LGALS9, which modulate T and B cell responses and support Treg differentiation through TGF-b), and triggering receptor expressed on myeloid cells 2 (TREM2, another protein that functions in immune response) are upregulated by uMφs. This fosters an anti-inflammatory environment that is crucial for the scarless regeneration of endometrial tissue during the proliferative phase [14]. These immune cells and their modulators undergo dynamic shifts to maintain endometrial homeostasis. Disruptions in their populations or functions can create an inhospitable environment for embryo implantation and potentially lead to endometrial pathologies like endometriosis. Exploring these intricate interactions sparks fascinating questions about the immune system’s influence on reproductive health [15,16].

## 4. Immune Cells in Endometriosis with a Comparison to Typical Human Endometrium

The information above underscores the critical role of immune cells in the human endometrium for maintaining tissue homeostasis and regeneration throughout the menstrual cycle. In endometriosis, often described as ‘endometrium-like tissue outside the endometrium’, these immune cells exhibit significant alterations in distribution and signaling pathways, contributing to both the initiation and progression of the condition. Notably, women with early-stage endometriosis exhibit elevated pro-inflammatory cytokines in their endometrial immune cells, raising intriguing questions about their role in disease development [7].

In addition to endometriotic tissue, the “eutopic endometrium” in patients with endometriosis shows significant hormonal and immune alterations, including elevated levels of ERα. By enhancing estrogen activity, this leads to abnormal proliferation and disrupts normal function, particularly when tissue is displaced during retrograde menstruation. Elevated ERα levels not only aid the adhesion and survival of endometriotic lesions but also promote inflammation and angiogenesis [17].

Moreover, progesterone resistance in the eutopic endometrium interferes with normal decidualization, potentially reducing receptivity to embryos. While significant changes in PRs (PR-B and PR-A) are not typically found in endometriotic lesions, mechanisms like promoter hypermethylation and microRNA dysregulation—particularly miR-29c, which affects FKBP4 expression—may explain the loss of PR-B and weakened P4 signaling, further contributing to this resistance [18,19].

The immune environment (Figure 2) in the eutopic endometrium of women with endometriosis differs notably from that of healthy women, particularly in the immune cell composition and activation of inflammatory pathways [20]. However, the precise proportions and distribution of immune cell subtypes in their eutopic endometria remain debated. These variations contribute to secondary epigenetic changes that can alter critical biological pathways. In the eutopic endometria of women with endometriosis, a defect in the suppressive function of non-T regulatory (non-Treg) cells, combined with the polarization of T helper 17 (Th17) cells driven by activated DCs, may promote a shift toward pro-inflammatory M1φs. This shift is further enhanced by elevated interleukin-17 (IL-17) levels, which are known to stimulate cytotoxic CD8+ T cells and aberrantly activate NKs. The IL-17 signaling pathway is significantly enriched in the eutopic endometrium of endometriosis patients, characterized by increased CD8+ T cells, NK cells, and follicular T helper cells compared to healthy controls. Additionally, the IL-17-producing non-Treg subset may contribute to the observed B cell enrichment in endometriosis, highlighting the complex interactions among different immune cell types [21,22,23]. Studies show that women with endometriosis have increased populations of CD8+ T cells and CD56+ NK cells, alongside a decrease in CD163+ Mφs in their eutopic endometrium. This shift reinforces a pro-inflammatory immune environment, with elevated CD8+ T cells potentially increasing the risk of infertility associated with the condition [22].

A recent study on cell clusters from menstrual effluent rich in endometrial tissue revealed striking differences between endometriosis patients and healthy controls during the menstrual phase. Endometriosis patients showed a significant depletion of uNK cells and an unexpected increase in B cells. Additionally, endometrial stromal cells from women with endometriosis exhibited a deficiency in P4-sensitive markers linked to decidualization. These key markers include insulin-like growth factor-binding protein 1 (IGFBP1, also known as placental protein 12, which binds insulin-like growth factors in plasma to prolong their half-life and alter interactions with surface receptors to produce different transcriptional splice variants), left–right determination factor 2 (LEFTY2, a member of the TGF-β family that plays a role in the developmental asymmetry and which is also implicated in endometrial bleeding), lumican (LUM, an extracellular matrix protein for collagen fibril organization, epithelial cell migration, and tissue repair), and decorin (DCN, a proteoglycan that modulates TGF-β and autophagy, and inhibits angiogenesis), all of which play important roles in tissue function and repair [25].

Mφs, vital for pathogen defense, also show significant differences in activity within the eutopic endometrium of women with endometriosis compared to healthy individuals, and recent single-cell studies indicate a pronounced pro-inflammatory phenotype among Mφs in endometriosis patients [7,14,26,27]. The predominant Mφ population is Mφ1 in these patients, which contributes to a detrimental inflammatory milieu affecting embryo implantation and endometrial functionalis layer regeneration. Interestingly, Mφs and DCs do not increase during the secretory phase, allowing cellular debris to accumulate and potentially migrate to the peritoneal cavity, where they tend to implant, allowing for the growth of ectopic lesions [8]. Recent data also suggest that the populations of Mφ1 and Mφ2 are key players in creating the abnormal inflammatory environment in endometriosis. At the mRNA level, sustained WNT signaling and dysregulated insulin signaling have been observed in early/mid-stromal cell populations in endometriosis, aligning with previous reports of IGF2 downregulation and impaired WNT inhibition during the secretory phase. This indicates altered WNT signaling in stromal cells, which is essential for the differentiation of the glandular epithelium in response to P4 [14]. These immune dysfunctions in the eutopic endometrium not only hinder anti-inflammatory responses but also enable endometrial cells to survive and migrate to ectopic sites, paving the way for lesion development.

Novel perspectives have attributed the imbalance of immune cell populations in endometriosis to the improper expression of immune checkpoints (ICPs) and their ligands. ICPs function in maintaining self-tolerance and the modulation (initiation, duration, and magnitude) of immune responses of effector cells, with co-stimulatory and co-inhibitory signaling between antigen-presenting cells and effector cells, a possible source of immune dysregulation in endometriosis [28]. These complex interactions highlight the urgent need for the further investigation into how immune and signaling pathways intertwine in endometriosis, potentially opening new avenues for targeted therapies that could restore normal immune function and improve reproductive outcomes.

Although our primary focus is endometriosis, it is important to also address adenomyosis—endometriosis located within the uterine myometrial muscles. Previous studies have underscored the crucial role of the immune profile in the development and progression of adenomyosis. The eutopic endometrium in adenomyosis exhibits immune-related alterations that distinguish it from the endometrial tissue of individuals without the disease. These changes contribute to a pro-inflammatory microenvironment characterized by impaired immune surveillance and dysregulated immune responses, similar to those of endometriosis. In particular, immune cells, such as Mφs and NK cells, display altered functionality, resulting in a reduced ability to clear abnormal cells. Additionally, the cytokine profiles in the eutopic endometrium reveal elevated levels of pro-inflammatory mediators, which further contribute to chronic inflammation and tissue remodeling. This immune dysfunction likely facilitates the migration of basal endometrial cells into the myometrium, promoting the progression of adenomyosis.

Dysregulated immune responses in the peritoneal environment may fail to eliminate misplaced endometrial cells, allowing them to implant [29,30]. Endometriotic cells that implant superficially on the peritoneum can proliferate and thrive in the peritoneal fluid, a specialized environment rich in estrogens, progestins, cytokines, growth factors, and other bioactive molecules. This setting supports the survival, growth, and invasion of ectopic endometrial cells, leading to chronic inflammation, neovascularization, and the progression of peritoneal endometriosis lesions [31].

Additionally, hormonal fluctuations during the menstrual cycle significantly alter steroid hormone levels in the peritoneal fluid, influencing the local microenvironment and potentially impacting the progression and development of endometriosis. Notably, during the follicular phase, E2 and P4 levels in the peritoneal cavity are significantly elevated compared to plasma, with P4 levels even reaching those observed during the luteal phase. At ovulation, the ruptured follicle releases steroid hormones into the peritoneal cavity, resulting in hormone concentrations that can be 100 to 1000 times higher than in plasma [29,32,33]. Ovulation also triggers a cascade of inflammation and resultant cell damage. Superficial peritoneal implants on the ovarian cortex might affect the ovulation site by altering the surrounding inflammatory microenvironment [34]. Key players in this process are cytokines such as interleukin-1, interleukin-6, and tumor necrosis factor, which are vital in both the inflammatory response associated with endometriosis and the mechanisms governing ovulation. This interplay raises intriguing questions about how these factors shape reproductive health [35].

In women with endometriosis, NK cells of the innate immune system appear to play a crucial role in the impaired clearance of endometriotic tissue within the peritoneal cavity. While there is some disagreement among studies regarding the quantitative alterations of peritoneal NK cells, a consensus has emerged that NK cells from both peripheral and peritoneal sources in endometriosis patients exhibit significantly reduced cytotoxic function. This diminished activity is closely linked to an increased expression of killer inhibitory receptors, which interact with HLA class I molecules on target cells, ultimately suppressing NK cell activity and function [36,37]. Compounding this issue, the cytotoxic function of CD8+ T lymphocytes in peritoneal fluid is also impaired, allowing endometriotic cells to evade immune surveillance, which is achieved by inducing T lymphocyte apoptosis through the Fas–FasL pathway [38]. Previous research has shown that endometriotic cells not only develop resistance to Fas-mediated NK apoptosis but also exploit this mechanism to launch a “Fas counterattack” against the host immune response. By upregulating FasL expression, these endometriotic cells induce apoptosis in immune cells, effectively shielding themselves from Fas-mediated cell death and enhancing their ability to evade immune detection, providing another nuanced interaction between immune evasion and tissue persistence, highlighting the complex nature of the underlying mechanisms at play in endometriosis [39].

In a prior study, researchers delved into what they referred to as endometriosis-associated immune cell infiltrates (EMaICIs) across various forms of endometriosis (including peritoneal–ovarian) and adenomyosis, and found that EMaICIs were most abundant in peritoneal and ovarian endometriosis, while adenomyosis exhibited a significantly lower presence of these infiltrates. The analysis identified key immune cell types, predominantly CD3+ T lymphocytes, which included both helper (CD4+) and cytotoxic (CD8+) T cells, as well as antigen-experienced memory T cells (CD45RO+). Additionally, Mφs and B lymphocytes were part of the infiltrate. Interestingly, regulatory T cells (CD25+ high and FoxP3+) and NKs (CD56+) were significantly absent [40].

Conversely, some studies suggest that the immune response action may be a protective mechanism that limits disease progression. Based on an investigation of immune activation in lymph nodes associated with the bowel lesions of deep endometriosis (DIE), lymph nodes showed increased T cell proliferation, particularly rich in CD4+, along with a diverse array of paracortical immune cells, including helper and regulatory T cells, B cells, DCs, Mφs, and plasma cells. The lymph nodes in question exhibited a notable decrease in CD10+ endometrial–stromal-like cells compared to other pelvic nodes. This observation suggests that endometriotic lesions or the movement of endometrial–stromal-like cells may trigger and activate the immune response, which drain them, potentially playing a role in controlling the disease’s spread and progression [41]. However, it is crucial to acknowledge the significant variability in tissue localization and the stages of endometriosis among the studies mentioned. This variability complicates the efforts to draw definitive conclusions, as many hormonal and environmental factors, along with mediators of pathogenesis, remain poorly understood and could influence these complex interactions.

## 5. Estrogen, Isoforms, and Their Transcriptional Regulation

Estrogens are a group of structurally similar steroid hormones derived from cholesterol, including estrone (E1), 17β-estradiol (E2), estriol (E3), estetrol (E4), and 27-hydroxycholesterol (27HC). E2 is the most potent one, with the highest affinity for estrogen receptors (ERs), and plays a crucial role in various biological processes [42,43]. E1 predominates in postmenopausal women, produced by gonads and adipose tissue, while estriol and E4 are primarily present during pregnancy. E2 exerts its effects through ER binding, regulating the cellular metabolism, proliferation, ion transport, and contractility [44].

Estrogen receptors, ERα and ERβ, encoded by the *ESR1* and *ESR2* genes, can form homo- or heterodimers with distinct functional roles. Their primary difference lies in the N-terminal domains, specifically the ‘activation-function 1 (AF-1)’ and the ‘ligand-binding domain (AF-2)’, which exhibit low sequence similarity (Figure 3). This affects their recruitment of coregulators and gene transcription regulation, with ERα demonstrating greater transcriptional activity due to its more potent AF-1 [45,46]. In the absence of E2, ERs remain inactive, bound to heat-shock proteins (HSPs) in the cytoplasm or nucleus. E2 binding induces a conformational change, releasing HSPs, dimerizing the receptors, and facilitating their translocation to the nucleus to interact with estrogen-response elements (EREs) in target gene promoters [47,48].

E2 also activates the non-genomic signaling pathway by binding to membrane-bound ERα and ERβ. This rapidly triggers nuclear transcription factors through ion channel regulation and the activation of enzymes like Ca^2+^ mobilization, PI3K, and MAPK. This process occurs within seconds to minutes, independent of gene regulation, and is known as the rapid “non-genomic effect” [49].

ERα and ERβ, the primary nuclear receptors for estrogens, are encoded by separate genes and perform various roles that can overlap, contrast, or act independently, depending on the cell type they are found in. ERα is more prevalent and is predominantly found in immune cells. In contrast, a definitive understanding of ERβ expression remains elusive, as many previous studies have been called into question due to issues with the accuracy of the antibodies used for detecting ERβ in cells [46,50].

These receptors are vital in regulating the bone metabolism, cardiovascular health, and central nervous system functions. Accordingly, the dysregulation of estrogen signaling may be associated with various diseases, including cancer and metabolic disorders. ERα expression typically increases in early breast cancer stages, facilitating tumor growth, and antiestrogens are often employed in therapies. Conversely, ERβ expression generally declines during carcinogenesis, suggesting a suppressive role and the potential as a therapeutic target. However, conflicting evidence regarding ERβ’s role in cancer arises from patient variability, tissue heterogeneity, and weak correlations between ERβ mRNA and protein levels, highlighting the need for further research to better elucidate its function in cancer as well [51].

Additionally, ‘G protein-coupled estrogen receptor 1 (GPER1)’, previously known as ‘G protein-coupled receptor 30 (GPR30)’, has emerged as another ER gene. Studies have shown that E2 can activate rapid signaling through the GPER1 protein. GPER1 has a unique pattern of expression in endometriosis. A study consisting of 74 ovarian, peritoneal, and deep infiltrating endometriosis showed high cytoplasmic GPER1 expression levels in the epithelial component of endometriosis and none in the normal endometrium. There was no significant difference in nuclear epithelial GPER1. The endometrioma has the highest frequency of cytoplasmic epithelial GPER1 as compared to peritoneal or deep infiltrating endometriotic lesions. GPER overexpression in endometriosis suggests roles in hormonal regulation. The inhibition of GPER in ectopic endometrial stromal cells using miRNAs reduced the proliferation, migration, and invasion [24,52,53]. These data suggest the importance of GPER in endometriosis progression.

## 6. Estrogen–Immunity Relationship and Alterations in Endometriosis

The immune system exhibits notable sexual dimorphism, with females generally displaying stronger immune responses and a higher incidence of autoimmune diseases than males. This disparity is linked to differences in estrogen levels and their effects [54]. Research has established a connection between fluctuations in circulating estrogen and the onset and severity of various autoimmune diseases, highlighting estrogens’ role as key immune modulators [55,56].

Specifically, E2 plays a crucial role in shaping the development and function of immune cells, as summarized in Table 1. Specifically, E2 is essential for the self-renewal of hematopoietic stem cells (HSCs) and further impacts the quantity and type of immune cells involved in both innate and adaptive immune responses, with studies demonstrating that ERα and ERβ mRNAs and proteins are expressed in hematopoietic progenitors and mature immune cells [44,57].

*ESR1* expression is the highest in B cells, with moderate levels in helper T cells, cytotoxic T cells, NK cells, and plasmacytoid DCs. Interestingly, while monocytes exhibit the lowest *ESR1* RNA levels, this expression increases in monocyte-derived DCs during differentiation. *ESR2* is primarily expressed in B cells and plasmacytoid DCs, while *GPER1* is also found in various immune cells, including T cells, B cells, monocytes, Mφs, and neutrophils [57,58,59,60]. Moreover, estrogen protects primary B cells from apoptosis by repressing PD-1 and promotes the differentiation, proliferation, and survival of early B-lineage precursors by upregulating CD22, a transmembrane protein that regulates B cell function; SHP1, a protein tyrosine phosphatase expressed in all hematopoietic cells and a key regulator of immune cell function; and Bcl-2, the protein which controls cell fate by blocking apoptosis [61,62]. These findings suggest that estrogens modulate immune responses based on ER expression, and can significantly impact the differentiation and function of neutrophils, Mφs, and NK cells [57].

In endometriosis, estrogen significantly impacts both immune responses and disease progression. ERβ plays a crucial role in mediating E2 effects, supporting the survival of endometriotic lesions, promoting pelvic peritoneum remodeling, and inducing inflammatory mediator release. A unique known challenge in endometriosis, P4 resistance, stems from the deficiency of P4 in the stromal cells of endometriotic tissue, complicating the hormonal and immune interplay characteristic of the condition [63,64].

The expression levels of ERs vary markedly between the healthy endometrium and ectopic lesions or the eutopic endometrium. In endometriotic stromal cells, ERβ is significantly overexpressed, while ERα levels are notably lower compared to endometrial stromal cells of the healthy endometrium. This receptor imbalance is linked to endometriosis pathogenesis and is influenced by aberrant epigenetic modifications, such as DNA methylation in endometriotic cells. Despite the considerable data on ERs in endometriosis, the specific contributions of ERα and ERβ to disease development remain inadequately understood [65]. In particular, the ERα distribution in superficial peritoneal lesions shows significant variability, even within single biopsy samples [66]. This heterogeneity also obscures the influence of menstrual cycle fluctuations on receptor expression, complicating the accurate classification of the ER status in the disease.

The abnormal activity of ERs, particularly their elevated expression in localized endometriotic tissues, likely drives their growth and persistence. Additionally, local de novo estrogen production within endometriotic tissues further disrupts the microenvironment, altering steroid hormone dynamics and triggering secondary immune responses. These shifts in the tissue microenvironment and immune activity are critical to the development and persistence of endometriosis [67]. It is also important to recognize that these hormone receptors may be upregulated in response to hormonal fluctuations or tissue-specific signals. While their roles under physiological conditions are not fully elucidated, understanding how they affect immune cells is essential, prompting further investigation.

**Table 1 cells-14-00058-t001:** The role of estrogen on immune cell function in the human endometrium and endometriosis.

Immune Cell	Immune Cell Phenotype Marker	Estrogenic Effect on Immune Cell	General Role of Immune Cell in Healthy Endometrium	Role of Immune Cell in Endometriotic Lesions	Refs.
Macrophages (Mφ1s)	CD14+, CD68+, HLA-DR+, CD11c−, and CD86+, CD80+, or iNOS+ (pro-inflammatory)	17β-estradiol significantly impairs the gene expression of inflammatory markers and the production of IL-1β.	- Proliferative: Mφ1 activity is balanced with Mφ2, maintaining immune tolerance and preparing for potential implantation. - Secretory: Mφ1 cells help in controlling inflammatory response. - Menses: Mφ1 cells dominate to Mφ2, and contribute to tissue shedding by increasing the activity through estrogen drop.	Persistent Mφ1s exacerbate inflammation in lesions and may interfere with effective lesion clearance.	[68,69,70,71,72]
Macrophages (Mφ2s)	CD14+, CD68+, HLA-DR+, CD11c−, and CD163+ or CD206+ (anti-inflammatory)	17β-estradiol significantly upregulates the expression of anti-inflammatory markers and enhances migration.	- Proliferative: High estrogen shifts the phenotype towards Mφ2 polarization, promoting tissue repair and immune tolerance. Increased activity supports endometrial regeneration. - Secretory: The Mφ 1 phenotype dominates to Mφ1. Gland remodeling and angiogenesis through VEGF secretion, and function in tissue preparation for implantation. - Menses: Mφ2 activity increases, reducing inflammatory responses.	Reduction in the wound-healing phenotype CD163+. Persistent Mφ2 activity supports immune tolerance in lesions.	[71,72]
Neutrophils	CD16+ and CD66b+ or CD15/SSEA1+	17β-estradiol; reduced adhesion (CD11b, CD18, and CD62L); selective inhibition of phagocytosis; E2 concentration-dependent suppression of ROS. High E2 levels increase NET formation, enhancing inflammation.	- Proliferative: Elevated estrogen increases neutrophil influx, with a moderate inflammatory role. - Secretory: Estrogen suppresses neutrophil activity, aiding in tissue maintenance and avoiding excessive inflammation. - Menses: Neutrophils contribute to tissue shedding and the clearance of cell debris through neutrophil extracellular trap (NET) formation.	Increased sensitivity to estrogen, leading to impaired immune clearance and exacerbated tissue damage, and inflammation around the lesions. Increase in NET formation may exacerbate local inflammation, contributing to lesion persistence.	[73,74,75,76]
Eosinophils	CD11b+, CD16+, CD193+, and Siglec-8+, and CD16−	Primarily enacted through Erα; GPER increases CCL1 chemotaxis, modulates apoptosis (blocks caspase3), and increases IL-5-stimulated apoptosis.	- Proliferative/Secretory: Eosinophils are not prominent. - Menses: Eosinophil numbers increase, facilitating tissue breakdown during menstruation. GPER aids in tissue remodeling through blocking caspase-3 and enhancing chemotaxis.	Eosinophils are often decreased, and they exhibit altered function, which may contribute to the failure in immune surveillance and inflammation.	[77,78]
NK Cells	CD56low, CD16+; CD56+, NCR+, and CD69+	Suppresses NK cell cytotoxicity.	- Proliferative: Estrogen supports NK cell function, promoting activation and immune surveillance, and aiding in tissue homeostasis. - Secretory: Estrogen reduces NK cytotoxicity, and promotes secretion of factors to maintain uterine vascular stability and tolerance. - Menses: NK cell cytotoxic activity peaks to assist with tissue breakdown.	NK cytotoxicity declines as the disease progresses, supporting lesion persistence and inflammation.	[79,80,81]
T Cells (CD8+ Cytotoxic)	CD3+, CD8+, Granzyme+, or Perforin+	Enhances effector functions (cytokine secretion and lytic activity) predominantly through Erα, with Erβ having a more inhibitory effect.	- Proliferative: Cytotoxic T cell activity is downregulated by estrogen, promoting tissue growth. - Secretory: Cytotoxic T cell function further suppressed to avoid damaging the endometrium. - Menses: Cytotoxicity increases to aid in shedding and tissue breakdown.	Cytotoxic T cell activity is low in lesions, promoting immune tolerance and the persistence of ectopic tissue.	[82,83,84]
T Cells (CD4+ Helper)	CD4+, T-Bet+, and IFNγ+; GATA-3+ and IL-4+; RORγt+ and IL-17+; FoxP3+ and CD25+	Supports the balance between Th1 and Th2, where elevated E2 favors Th2 in human unstimulated PBMCs.	- Proliferative: Estrogen supports a balance between Th1 and Th2 cells for tissue homeostasis. - Secretory: Higher estrogen promotes Th2 differentiation, which suppresses inflammatory responses, which supports embryo implantation. - Menses: Low estrogen promotes Th1 differentiation, allowing for inflammation and immune activation processes. Th17 cell activity increases, potentially aiding in uterine shedding.	Th1/Th17 cells drive inflammation in lesions, contributing to disease pathogenesis. Their persistence in lesions may exacerbate tissue damage and chronic inflammation.	[83,85]
Regulatory T Cells (Tregs)	CD4+ and FoxP3+	Elevation influences Treg expansion, promoting immune tolerance and tissue homeostasis.	- Proliferative: Tregs help suppress excessive immune responses by regulating Th1/Th2. - Secretory: Tregs promote immune tolerance by secreting immunosuppressive cytokines (IL-10 and TGF-β), preventing inflammation to support implantation. - Menses: Reduced Treg function increases inflammation but prevents excessive immune activation.	Tregs are often dysregulated in lesions, contributing to the persistence of inflammation and lesion survival. Estrogen influences Treg expansion, supporting lesion survival.	[21,83,86]
B Cells	CD19+	Impacts B cell differentiation, survival, and function primarily by upregulating CD22, SHP-1, and BCL-2.	- Proliferative: Lower estrogen stimulates B cell differentiation, allowing for the secretion of antibodies and cytokines for the regulation of tolerance and defense in tissue. - Secretory: Estrogen upregulates CD22, SHP-1, and BCL-2 factors, enhancing B cell survival, and conferring tolerance during embryo implantation. - Menses: Estrogen levels decrease, altering B cell populations; reduction in cell survival factors, causing an increase in immature B cell populations and the apoptosis of Breg cells.	Estrogen suppresses B cell activity in lesions. Estrogen may promote autoreactive B cell survival, contributing to inflammation.	[62,87]
Dendritic Cells (DCs)	CD11c+, HLA-DR+, and CD80/86+	17β-estradiol inhibits DC maturation, reducing antigen presentation.	- Proliferative: DCs assist in the immune response to potential pathogens. - Secretory: Reduced DC maturation, reducing the antigen presentation capacity and supporting immune tolerance. - Menses: DCs modulate local immune activity and prevent excessive inflammation, supporting tissue turnover.	Reduced DC activity aids in lesion survival. Estrogen modulates DC function, promoting the immune tolerance of lesions. DCs might increase inflammation, supporting lesion breakdown.	[88]
Conventional DC1s	CD11c+, HLA-DR+, and XCR1 or CLEC9α+	Promotes the differentiation of cDC1 but reduces cDC1 maturation, impairing antigen presentation.	- Proliferative: Differentiation is promoted to aid in antigen presentation, immune surveillance, and priming T cell responses. - Secretory: Suppression of inflammation and inducement of immune tolerance during implantation. - Menses: DCs help to promote shedding.	DC1s are less active in lesions, with impaired antigen presentation and immune responses. This promotes immune suppression in lesions, aiding in tissue survival and persistence.	[88]
Myeloid-Derived Suppressor Cells (MDSCs)	CD11b+ and Gr1+	Positive correlation between E2 and MDSC in a dose–response manner, resulting in expansion (*S100A8* and *S100A9* upregulation), functioning in suppressing T cells, and elevated ROS production.	- Proliferative: Elevation of E2 results in the expansion and activation of MDSCs to regulate immune responses to support endometrial tissue. - Secretory: MDSCs suppress inflammation through secreting anti-inflammatory cytokines (IL-10) and suppress T cell activity for a tolerogenic state to support embryo implantation. - Menses: Estrogen declines, allowing for a tolerogenic shift, allowing for controlled inflammation and the shedding of endometrial lining.	MDSCs promote immune suppression in lesions, aiding in tissue survival and persistence.	[89,90,91]
Mast Cells	CD117+ and FcεRI+	17β-estradiol increases the infiltration and differentiation of mast cells by upregulating CPA3, VCAM1, CCL2, CMA1, CCR1, and KITLG.	- Proliferative: Mast cells are involved in immune modulation through the release of cytokines, growth factors, and histamine, facilitating an early immune response for tissue remodeling. - Secretory: Mast cell factors are repressed reducing the inflammation of the endometrium and confer tolerance for embryo implantation. - Menses: The upregulation of inflammatory mediators to facilitate menstruation and immune response.	Increased, and believed to contribute to chronic inflammation, enhancing the growth and survival of lesions.	[5,78]

## 7. Immune–Endocrine Interactions in Endometriosis: Cellular Mechanisms and Pathogenic Pathways Through the Immune Cell Lineage

The development of endometriosis cannot be attributed to a single pathway, as it results from the interplay of multiple physiological systems, with the immune and endocrine systems playing particularly influential roles. Understanding the mechanisms of immune–endocrine crosstalk in endometriosis is crucial for improving patient outcomes and developing more effective therapeutic strategies. ERα directly impacts HSCs, lymphoid progenitors, and myeloid progenitors, promoting various developmental pathways. A key consequence of ER action in these progenitors is the potential induction of epigenetic changes in precursor cells, influencing both developmental pathways and the functional responses of mature cells. Given this complexity, our focus in this part of our review will be on immune cells, seeking to re-evaluate their individual roles within the endometriotic milieu to better understand the disease, as summarized in Figure 4.

### 7.1. Myeloid Lineage

The deletion of a conditional ERα allele specifically in myeloid cells has revealed that E2 enhances Toll-like receptor 4 (TLR4) signaling in Mφs in vivo, a critical component in the innate immune response, resulting in increased levels of IL-6, IL-1β, and inducible nitric oxide synthase (iNOS), which contribute to a pro-inflammatory response; E2 amplifies this pathway by suppressing PI3K signaling and reducing Akt phosphorylation in Mφs [92]. These findings, along with related studies, highlight the heightened immune response often seen in females, which may also play a role in the increased prevalence of inflammatory diseases [60].

GPER-E2 signaling also activates the PI3K pathway, enhancing cell survival through the Akt signaling cascade. This response helps cells to counteract the stress caused by endoplasmic reticulum dysfunction by enhancing protein folding and preventing apoptosis, thus facilitating proliferation and migration. The activation of the tyrosine kinase Src pathway relocates chaperone proteins to the cell surface, further promoting cell survival. These mechanisms culminate in the activation of heparin-bound EGF, which transactivates its receptor, EGFR. The subsequent EGFR activation stimulates both the PI3K/Akt and ERK/MAPK pathways, again contributing to the regulation of cell proliferation and migration [74,93,94].

### 7.2. Myeloid-Derived Suppressor Cells (MDSCs)

MDSCs are a diverse group of immature and mature myeloid cells, including precursors for DCs, Mφs, and granulocyte-like neutrophils, eosinophils, basophils, and mast cells (MCs). MDSCs interact with various lymphoid cell types, such as NK cells, and T and B lymphocytes, and are classified into the following three main subsets: monocytic (M-MDSCs), polymorphonuclear (PMN-MDSCs), and early stage (eMDSCs) [90,95].

The immunosuppressive functions of MDSCs are mediated through multiple mechanisms. They suppress T cell activity by inducing ARGinase-1 (ARG-1), which depletes arginine and impairs T cell activation. MDSCs also produce nitric oxide (NO) and reactive oxygen species (ROS) via iNOS, further inhibiting T cell function. Additionally, they secrete TGF-β, which not only inhibits T cell activation but also reduces the cytotoxicity of cytotoxic T cells and suppresses the differentiation of Th17 cells [90,96,97].

MDSCs play a pivotal role in creating an immunosuppressive microenvironment and are associated with poor prognosis in cancers. In cancer immunology, E2 has been shown to influence the differentiation of E2-insensitive tumors by modulating MDSCs and exerting a significant inhibitory effect through the inhibition of ERα [98]. Moreover, MDSCs were suggested to serve as biomarkers in endometriosis, where elevated levels correlate with peritoneal lesions in patients with advanced disease, reflecting a sustained immunosuppressed niche [99].

### 7.3. Macrophages

Mφs recruited to the uterus primarily originate from the bone marrow, responding to tissue-specific signals related to damage and inflammation. In healthy females, tissue-resident Mφs are found in the perimetrium, myometrium, and endometrium, with densities varying throughout the menstrual cycle [100]. During the proliferative phase, under the influence of an E2 surge, Mφs (expressing markers CD54+, CD69+, and CD71+) constitute 1–2% of endometrial cells, playing a key role in supporting tissue proliferation and regeneration [68]. In the secretory phase, Mφ numbers rise to 3–5%, contributing to gland remodeling and angiogenesis through VEGF secretion. During menstruation, Mφs constitute 6–15% of the endometrial cell population, releasing matrix metalloproteinases, including MMP-12, MMP-9, and MMP-14, which are essential for the breakdown of the functionalis layer [68,101]. Throughout these phases, Mφs facilitate the resolution of immune responses and restore tissue homeostasis after inflammation and remodeling.

Estrogen significantly influences Mφ function, although the expression of estrogen ERα and ERβ in these cells is debated [69,70,102,103]. Some studies indicate that E2 regulates Mφ activation and cholesterol homeostasis, induces alternative activation in vitro, and affects the expression of the AF1-deficient form of ERα (the so-called ERα46) in human Mφs [70]. Additionally, ERα deficiency has been linked to increased TNFα secretion in murine Mφ in response to microbial stimuli [104]. Chronic E2 administration in ovariectomized mice triggers a pro-inflammatory response in resident peritoneal Mφ, while E2 surges promote cutaneous repair responses by enhancing Bcl-2 expression through ERK phosphorylation. In contrast, prolonged E2 exposure leads to a pro-inflammatory Mφ response via PI3K phosphorylation [92,105,106].

When it comes to the eutopic endometrium of endometriosis patients, as mentioned above, there are fewer “wound-healing” marker CD163+ Mφs and increased CCL2, indicating greater monocyte influx. Pro-inflammatory M1 Mφs accumulate due to defective non-T regulatory cells and Th17 polarization [14,61,62,63].

This altered Mφ activity creates a harmful inflammatory environment that hinders embryo implantation and endometrial regeneration. Additionally, Mφs do not increase during the secretory phase, leading to debris accumulation that can promote ectopic lesions, emphasizing the important roles of Mφ1 and Mφ2 [101].

In endometriosis, ectopic lesions increase while blood monocyte counts remain unchanged in patients with endometriosis; there is a fractional increase in ectopic lesions compared to the eutopic endometrium. However, the number of Mφs in peritoneal fluid rises significantly, with populations classified into the following two states: Mφ1 (pro-inflammatory) and Mφ2 (anti-inflammatory). The shift toward Mφ2, often seen during infection, relies on an ERα-mediated pathway that promotes interleukin-4 (IL-4) secretion, enhancing Mφ2 differentiation while inhibiting NFκB signaling and nitric oxide production, thus suppressing Mφ1 activity. Mφ1 is characterized by CD40+/CD80+/CD86+/HLADR+ markers, while Mφ2 expresses CD163+/CD206+/CD204+ markers [107].

Mφ polarization is influenced by various stimuli: Mφ1 develops from interferon-gamma (IFN-γ), TNF-α, or lipopolysaccharides, while Mφ2 is driven by IL-4, IL-10, IL-13, or TGF-β [107]. As both subsets increase regardless of the disease stage, the predominant Mφ phenotype in endometriosis remains debated, with some studies suggesting an Mφ1 dominance that may hinder embryo implantation [107,108]. However, the relationship between endometriosis and endometrial implantation may be the subject of another separate review [109].

The current understanding depicts Mφ polarization in endometriosis as dynamic, shifting from a pro-inflammatory Mφ1 state in early disease to an immune-suppressive-Mφ2 state in advanced stages [110]. Early ectopic endometrial cells secrete monocyte chemotactic protein 1 (MCP-1), recruiting additional Mφs. These local Mφs release TNF-α, IL-1β, IL-6, and MCP-1, promoting Mφ1 differentiation. As the lesions progress, additional factors drive the transition to the Mφ2 phenotype, facilitating angiogenesis, neurogenesis, and invasive lesion characteristics [8,111].

Based on research by Burns et al., early endometriosis development is primarily driven by immune responses, with minimal involvement of E2 and ERα. Significant infiltration of neutrophils and Mφs was observed, along with a reduced IL-6 response in lesions lacking ERα, suggesting two phases of the disease, an immune-dependent phase followed by a hormone-dependent phase, indicating that targeting the immune system may help prevent lesion attachment linked to Mφ activity [112].

### 7.4. Neutrophils

Neutrophils play a crucial role in the early progression of ectopic endometriotic lesions, with their recruitment driven by factors such as IL-8, human neutrophil peptides 1-3 (HNP1-3), and the Epithelial Neutrophil-Activating peptide, ENA-78 [30,113]. E2 not only increases neutrophil counts but also enhances their inflammatory activity by promoting the secretion of pro-inflammatory cytokines, including CXCL-10, IL-8, and VEGF [75]. E2 and its receptors regulate neutrophil function. Both ERα and ERβ are upregulated in pre-menopausal women during ovulation, coinciding with elevated serum E2 levels [73].

As mentioned above, neutrophils express GPER1, which interacts with key signaling pathways, including cAMP/Protein Kinase A (PKA), involved in regulating cell growth, migration, invasion, and metabolism, p38 Mitogen-Activated Protein Kinase (p38 MAPK), which coordinates cellular responses to stress and inflammation, and Extracellular Signal-Regulated Kinase (ERK), which has role in proliferation, differentiation, apoptosis, and survival. Activating GPER1 in neutrophils with the nonsteroidal agonist G1 promotes pro-inflammatory differentiation, prolongs lifespan, enhances respiratory burst activity, and modifies gene expression. Rodenas et al. propose that targeting GPER1 experimentally helps to reduce this pro-inflammatory neutrophil state [74].

In endometriosis, MyeloPerOxidase (MPO) levels are significantly elevated during the proliferative phase. Ectopic endometriotic lesions show the significantly increased mRNA expression of various chemokines such as CCL2, CXCL2, CXCL3, and CCL5; cytokines such as TNF, IL-16, and IL-7; Damage-Associated Molecular Patterns (DAMPs) such as S100A9; cell adhesion molecules like ITGAM, ITGB2, and SELP; and inflammasome markers such as NLRP3 compared to the eutopic endometrium and healthy controls. Furthermore, stage III/IV endometriosis lesions exhibit significantly higher levels of IL-8, a potent neutrophil chemoattractant, compared to controls [75].

### 7.5. Eosinophils

Eosinophils play a crucial role in the development and progression of endometriosis-associated fibrosis. Eosinophil peroxidase serves as an alternative marker for early-stage endometriosis-related fibrosis. Blumenthal et al. demonstrated that eosinophils co-localize with fibrotic connective tissues in endometriotic lesions, where they infiltrate and degranulate, potentially contributing to tissue remodeling and wound repair [114]. Furthermore, eosinophils enhance pro-tumorigenic processes by increasing the levels of VEGF, FGF, and PDGF, thereby promoting angiogenesis, tissue healing, and the secretion of extracellular matrix-degrading enzymes, while also inducing the Mφ2 phenotype through IL-13 secretion [115]. Most insights into E2-modulated eosinophil functions are derived from mouse models, with limited human studies. Evidence suggests that ERα is the primary mediator of E2 effects on eosinophils, facilitating their accumulation in the uterus; however, it remains uncertain whether this effect originates directly from eosinophils or eosinophil-responsive cells [116]. Eosinophils, initiators of inflammatory responses, are reported in an enriched manner in all phases of the cycle from the eutopic endometrium of endometriosis patients compared to the control endometrium, where they appear mainly pre- and during menses, to contribute to the pro-inflammatory phenotype [78,116].

### 7.6. Basophils

While IL-8, a chemoattractant for granulocytes, including basophils, is significantly elevated in endometriosis, literature confirming a substantial increase in basophil levels is limited [117]. However, a 2022 study by Bunis et al. employing whole tissue and single-cell RNA sequencing revealed a notable elevation of basophils in the eutopic endometrium of endometriotic patients throughout the uterine cycle, with these cells largely absent in healthy endometrial tissue.

### 7.7. Mast Cells

MCs, like Mϕs, are innate immune cells derived from common MDSCs with a unique ability to neutralize and degrade toxic proteins. Additionally, similar to Mϕs, MCs can adopt two distinct polarization profiles, which may serve opposing functions [101].

They are an integral part of the endometrial cycle, playing multifaceted roles in fibrosis, angiogenesis, inflammation, wound healing, and tissue remodeling. While the overall quantity of MCs relative to stromal cells remains relatively constant throughout the menstrual cycle, their morphology, granule composition, and activation states exhibit distinct variations at different stages. Just before menstruation, resident MCs become activated, facilitating tissue degradation through the ischemic vasospasm of spiral arteries via the release of heparin and histamine, and by enzymatically degrading the matrix in stromal fibroblasts. MCs are additionally involved in neuroinflammatory processes and pain perception, interacting with the neuropeptide substance P and the Mas-related G protein-coupled receptor (MRGPR). This interaction stimulates the release of histamine, IL-6, and TNFα [78,101,118]. The MRGPR family of receptors is specifically characterized by unique expression profiles and a range of ligands to br, which serve as crucial players in inflammatory responses and present potential therapeutic targets for various conditions. Immune cell-specific MRGPRs have been implicated in modulating adverse drug reactions, inflammatory diseases, bacterial defense mechanisms, and reactions to environmental irritants [119].

In the setting of endometriosis, MCs are associated with pain and are contributors to lesion fibrosis, angiogenesis, and cellular proliferation. Their activation correlates with the formation of fibrous adhesions and the upregulation of genes related to MC activity. Endometriotic lesions are also marked by heightened levels of stem cell factor (SCF). Studies involving the inhibition of the MC/stem cell growth factor receptor KIT (cKIT) with pexidartinib—an agent that disrupts pathways linked to inflammation and cell survival—have shown a reduction in MC activation [120,121]. While cKIT expression remains consistent across the menstrual phases of the healthy endometrium, it is significantly elevated in peritoneal and ovarian endometriotic lesions. This suggests that elevated E2 levels in endometriosis promote MC infiltration and activation within the lesions, with SCF contributing to a positive feedback mechanism that enhances their recruitment and maturation [120,121].

### 7.8. Dendritic Cells

DCs arise from CD34+ hematopoietic stem cells in the bone marrow, maturing into the following two primary subsets: HLA-DR+/CD11c+ conventional DCs (cDCs) and HLA-DR+/CD123+ plasmacytoid DCs (pDCs). These versatile cells permeate various tissues and traverse the blood and lymphatic systems [122]. Paharkova-Vatchkova et al. demonstrated that E2 fosters the differentiation of functional DCs from murine bone marrow precursors, notably enhancing the intermediate CD11c(+) CD11b populations, which exhibit elevated MHC class II and CD86 expression [123]. This process is predominantly mediated by ERα, which is essential for maintaining normal DC function [99,117]. Additionally, IL-27 has been shown to upregulate *ESR1* gene expression in cDCs [88,124], while ERβ mRNA is also present, indicating that estrogen plays a significant role in their development and function [88,124].

Emerging research underscores the essential function of DCs throughout the menstrual cycle, across all phases, although their distribution remains contentious. Laginha et al. reported an increase in immature DCs (imDCs) during the proliferative and secretory phases of the healthy endometrium, with a subsequent decline during menstruation [122]. This initial influx is believed to facilitate embryonic implantation and support post-menstruation tissue repair by generating matrix metalloproteinases (MMPs), cytokines, and chemokines (IL-6, IL-10, IL-12, TNFα, and RANTES) [73]. In endometriosis, Schulke et al. reported pronounced imDCs during the proliferative and menstrual phases compared to controls, while mature DCs (mDCs) remained sparse [125]. Conversely, Lundberg et al. found no significant disparity in peritoneal fluid DCs between endometriosis and control samples; however, they reported an elevation in mannose receptor-positive (MR+) type 1 DCs, crucial for antigen recognition and Th2 differentiation, in endometriosis [126].

Furthermore, Izumi et al. indicated that a decrease in DC populations correlated with increased lesion burden and diminished T lymphocyte activation, suggesting that DCs in endometriosis may adopt a more phagocytic, tissue-clearing role. This highlights their potential in modulating immune responses and shaping the inflammatory landscape of endometriosis lesions, ultimately influencing disease progression [127]. Furthermore, E2 has role in altering DC behavior and cytokine production, potentially through neutrophil recruitment and activation, expanding the endometriosis–immunological dynamic complexity [35,128].

### 7.9. Platelets

Investigations have unveiled ERs on activated platelets and their precursor megakaryocytes, explaining how hormonal shifts might modulate platelet dynamics. Activated platelets significantly amplify E2 synthesis from endometriotic stromal cells through NF-κB and TGF-β1 pathways, facilitating the upregulation of pivotal hormone synthesis proteins like StAR, HSD3B2, aromatase, and HSD17B1, indicating a feedback loop that exacerbates endometriosis [129,130]. Additionally, platelets interact with endometriotic lesions, particularly in highly vascularized regions, where they become activated and cluster around VEGF expression [131].

Based on in vitro co-culture experiments, platelets not only boost the proliferation of stromal cells in endometriosis, but also elevate pro-angiogenic factors, including VEGF, COX-2, and MMP-9, and the anti-apoptotic protein Bcl-2 [88]. The critical involvement of platelets was further highlighted by in vivo studies showing that platelet depletion markedly reduced the lesion size and alleviated hyperalgesia in a mouse endometriosis model [132]. Additionally, Ding et al. reported increased markers of coagulation and platelet activity, as reflected by elevated surface P-selectin, enhanced platelet aggregation, higher plasma D-dimer, fibrinogen, fibrin degradation products (FDPs), soluble P-selectin, and a shorter thrombin time, signifying faster hemostasis in patients with ovarian endometriomas [133].

### 7.10. Lymphoid Cells

Lymphocytes are essential to the adaptive immune response, consisting of T cells, B cells, and NKs. T cells include helper T cells (CD4+), which secrete cytokines, and cytotoxic T cells (CD8+), which eliminate infected cells. B cells, derived from the bone marrow, produce antibodies. Both T and B cells generate memory cells that “remember” previous antigens, enabling a rapid response to future infections, a key aspect of acquired immunity.

Lymphocytes are pivotal, not only in immunity, but also in autoimmune disorders, facilitating the proliferation of autoreactive T and B cells, which can result in organ damage. Their sex hormone receptor expression induces nuanced epigenetic and transcriptional alterations that delicately modulate immune responses by activating immune-related genes, either directly or indirectly [134]. In the context of autoimmunity, E2 downregulates the AutoImmuneREgulator (AIRE, a transcription factor expressed in the medulla of the thymus) via DNA methylation, diminishing the expression of tissue-specific antigens and hindering Treg differentiation. This allows self-reactive T cells to evade negative selection, increasing autoimmune vulnerability in females [135].

Estrogen’s influence on T cell populations is dose-dependent: low levels promote CD4+ T cell expansion, while elevated concentrations, such as those during pregnancy, lead to a reduction in both CD4 and CD8 T cell counts [136,137]. Thymocytes also express ERs, and high levels of exogenous E2 decrease thymic cellularity, resulting in thymic atrophy [138].

### 7.11. Cytotoxic T Cells (CD8+)

Cytotoxic (CD8+) T cells are essential for adaptive immunity and immune surveillance, with ERβ mRNA expression. In the human endometrium, E2 inhibits the secretion of perforin and granzymes, diminishing the cytotoxic function of cytotoxic T cells [139]. This regulation is vital for reproductive success, as their suppression cells foster a tolerogenic environment that supports implantation and conception

In endometriosis, cytotoxic T cells are more prevalent within endometriotic lesions, yet no significant differences are found in peripheral blood. However, the presence of these cells in peritoneal fluid and eutopic endometrium remains inconsistent [140]. To enhance our understanding of eutopic immune cells, two studies explored immune cell variability in menstrual effluent, revealing no significant differences in CD8+ or CD4+ T cell populations, or their ratio. Interestingly, they reported a decrease in perforin high-expressing cytotoxic T cells [141]. Additionally, Shih et al. reported the identification of four unique cytotoxic T cell clusters, without further characterization [25].

### 7.12. CD4(+) T Helper (Th1 and 2) Cells

In the human eutopic endometrium, CD4+ T helper (Th)1 and 2 coexist, with their balance influenced by E2 levels. Low E2 levels favor the Th1 phenotype, associated with cell-mediated immunity and inflammation, while higher physiological doses promote the Th2 phenotype, linked to antibody production and anti-inflammatory responses [56]. These pathways inhibit each other: Th1 is characterized by IFNγ, which stimulates pro-inflammatory cytokines like IL-1β and TNF-α, whereas Th2 is marked by IL-4, inducing anti-inflammatory cytokines such as IL-10 and TGF-β [85]. In endometriosis, the immune environment is shaped by both Th1- and Th2-related inflammatory cytokines, although the precise effects of Th1/Th2 differentiation on the condition is not yet clear. Research shows that various signaling pathways regulate cytokines, fostering immune tolerance within ectopic lesions. In advanced stages, the Th2 response predominates, elevating the levels of IL-4, IL-6, and IL-13, while Th1 cytokines persist throughout the disease [142].

Gonadotropin-releasing hormone agonists (GnRH-as) competitively bind to pituitary GnRH receptors, leading to increased TNF-α and decreased IL-10 levels in vitro, thus raising the Th1/Th2 ratio. Beyond alleviating pelvic pain, studies suggest that GnRH-a may inhibit cell proliferation and suppress inflammation in the peritoneal environment [143]. The timing and mechanisms underlying Th2 dominance warrant further exploration. Future therapies may aim to target these cellular and molecular pathways, restoring normal immune regulation and overcoming immune tolerance, potentially revolutionizing endometriosis treatment.

### 7.13. Th17

These cells are a subset of T cells that produce cytokines such as IL-17, GM-CSF, and IFNγ, playing a crucial role in immune responses and tissue inflammation, particularly against extracellular bacterial and fungal pathogens. Studies indicate a strong correlation between elevated Th17 cells, along with increased IL-17 in peritoneal fluid and worsened inflammatory profiles, as well as increased disease severity in women with endometriosis. These cells release a range of cytokines, including IL-17A, IL-22, and IL-21, which are integral to host defense, autoimmunity, inflammation, and tumor development. In the context of endometriosis, Th17 cells not only promote cell survival by conferring a resistance to NK cell cytotoxicity through of ERK1/2 signal activation, but IL-17 and IL-10 also simultaneously drive angiogenesis and the perpetuation of inflammatory processes [21,144,145,146].

### 7.14. Treg

Regulatory T cells (Tregs), once known as suppressor T cells, play a crucial role in modulating the immune system and maintaining self-tolerance, which helps to prevent autoimmune diseases. Functionally, Tregs act as immunosuppressors by suppressing effector T cells, and are characterized by the expression of CD4, FOXP3, and CD25, sharing a lineage with naïve CD4 cells [147].

Within the eutopic endometrium of endometriosis patients, Tregs help to suppress the immune system’s recognition of endometriotic cells, enabling these cells to evade immune clearance. The elevation of Tregs during the secretory phase is thought to contribute to the ability of ectopic cells to evade immune surveillance. While the healthy secretory endometrium typically shows an increased infiltration of effector immune cells, in endometriosis, the continued elevation of Tregs from the proliferative to the secretory phase may limit the ability of immune cells to target endometrial antigens during menstruation, thereby facilitating immune evasion and the persistence of ectopic lesions [148].

Research indicates that E2 plays a significant role in Treg dynamics, and encourages Treg differentiation by increasing IDO1 expression in ectopic tissues [149]. When combined with CD3+/CD28+ stimulation, E2 further boosts Treg expansion without compromising its suppressive function, as with what occurs in pregnancy [150]. Patients with sustained high Treg levels often face fertility challenges, aligning with the E2 dominance characteristic of endometriosis [108].

### 7.15. NK Cells

In vitro and in vivo analyses of uNK cell differentiation reveal that this process continuously occurs in response to endometrial regeneration, driven by IL-15. Differentiated uNK cells exhibit reduced proliferation and immunomodulatory functions, including enhanced angiogenesis. Additionally, the uNK cell niche can be replenished from the circulation, which is under genetic control, indicating that the differentiation of human NK cells in the uterus is linked to significant functional changes associated with local tissue regeneration [146].

In healthy women, peripheral NK (pNK) cell populations fluctuate significantly during the menstrual cycle and play a crucial role in menstruation [137]. However, studies in endometriosis present conflicting results regarding pNK cell numbers; some show no change or a slight decrease, while others report increases, particularly in advanced disease stages. Despite this variability, it is widely acknowledged that NK cell cytotoxicity diminishes in women with endometriosis, with the impairment becoming more pronounced in advanced stages. While pNK cells are more abundant in early-stage endometriosis, their cytotoxic function declines as the disease progresses. Research indicates a decrease in activating receptors like NKG2D and NKP46, alongside an increase in inhibitory receptors such as KIRs and LILRB1, which facilitate immune evasion by ectopic endometrial cells [81,113].

The interaction between NK cells and estrogen is critical in endometriosis pathogenesis. E2 suppresses NK activity, impairing immune surveillance and promoting the survival of ectopic lesions. Moreover, NK cells are involved in regulating endometrial receptivity, follicle development, and immune evasion, all of which may contribute to implantation failure and infertility in affected patients [151].

NK cell dysfunction significantly influences the adhesion, proliferation, and growth of endometriosis, a central aspect of its pathophysiology. While NK dysfunction is recognized as a major contributing factor, the specific relationships among NK cell activity, symptomatology, and various endometriosis subtypes remain poorly understood [152].

Although the percentage of NK cells in the peritoneal fluid, endometrium, and peripheral tissue compartments is similar in women with and without endometriosis, the cytotoxic activity of NK cells is notably reduced in affected individuals. This reduction is linked to altered receptor expression and cytokine modulation, contributing to immune evasion and the persistence of ectopic lesions, underscoring the importance of NK cell cytotoxicity in both the endometriosis pathophysiology and potential therapeutic approaches [152,153].

### 7.16. B Cells

In the healthy endometrium, B cells constitute less than 5% of the leukocyte population and less than 2% of total cells, primarily located in the basal and stromal compartments. In ectopic endometrial lesions, B cells account for 17% of leukocytes, compared to 12% in the eutopic endometrium and 10% in controls [154,155]. Despite being classified as mucosal tissue, B cells in the endometrium are understudied compared to their role in peritoneal fluid in endometriosis.

E2 plays a dual role in B cell dynamics by suppressing lymphopoiesis while enhancing B cell function through interactions with the estrogen receptors ERα and ERβ. Grimaldi et al. found that E2 upregulates the critical molecules in B cells, including CD22, SHP-1, VCAM, and BCL-2. The increased expression of CD22 and SHP-1 diminishes the receptor-mediated signaling, while higher levels of BCL-2, an anti-apoptotic factor, promote the survival of pro/pre- and immature B cells. Additionally, E2-induced SHP-2 and VCAM are linked to the survival of autoreactive B cells, potentially increasing the risk of autoimmunity in females [62]. This estrogenic influence on B cell development correlates with the immune tolerance observed in endometriosis.

## 8. Conclusions

Immune dysregulations play pivotal roles in the development and progression of endometriosis, although the precise molecular mechanisms underlying their dysregulations remain poorly understood. The hormonal imbalances in endometriosis, particularly the overexpression of aromatase and aberrant steroid biosynthesis, lead to an estrogen-dominant and progesterone-resistant microenvironment. This endocrine shift perturbs immune cell behaviors and function, yet the full extent of these interactions is still unclear.

Estrogen is a significant immunomodulator and affects many immune cell types. While studies have shown the presence of ERs in various immune cells, there is still uncertainty regarding their expression levels and functional significance in the context of endometriosis. The expression of ERs and PRs in tissue-resident immune cells within the endometriotic lesions remains poorly characterized. Additionally, immune cell subsets, their polarization, and activation states, for example, for Mφs, DCs, NKs, T helpers, Treg cells, and platelets, should be more closely studied. Epigenetic modifications affect ER and PR expression in endometriosis, adding another layer of complexity.

The exact pathways of how immune cells sense the estrogenic environment is not particularly clear. It is intriguing that immune cells may utilize alternative pathways to respond to hormonal changes in an estrogen-dominated environment, potentially involving non-genomic signaling pathways and GPER signaling. Understanding how these pathways differ to classical ER signaling, and their contributions to the differentiation and survival of endometrial-resident immune cell populations, require further investigation.

Despite these insights, there are several limitations that need to be considered. First, extrapolating the findings from in vitro studies to clinical practice remains a challenge, as laboratory conditions may not fully replicate the complexity of in vivo environments. Additionally, while hormonal imbalances are well-documented in endometriosis, the precise relationship between hormonal levels and their clinical effects—such as symptom severity and disease progression—remains unclear. Not only the heterogeneity of endometriosis itself, but also the heterogeneity of immune responses in endometriosis, further complicates the interpretation of results, as different subtypes of the disease may exhibit distinct immune signatures even in different phases of the disease. Moreover, the availability and accessibility of patient tissue samples limit the scope of some studies.

Finally, the applicability of current findings in clinical practice is limited by the lack of standardized protocols for sample collection, processing, and analysis, which hampers the reproducibility and comparison across studies. Moving forward, future research should aim to clarify the immunological heterogeneity of the disease, focusing on both the functional roles of immune cell subsets and the influence of hormones on their behavior. Additionally, studies should seek to integrate clinical data with molecular findings to better understand the relationship between immune dysregulation and disease symptoms. These efforts will be essential for developing more effective and personalized therapeutic strategies for endometriosis.

This review underscores the need for further well-planned collaborative multicenter research to elucidate the role of hormone receptors in immune cells and their interactions within the endometriotic microenvironment. These insights may pave the way for targeted therapeutic approaches aimed at modulating immune responses to alleviate the symptoms and progression of endometriosis.

## Figures and Tables

**Figure 1 cells-14-00058-f001:**
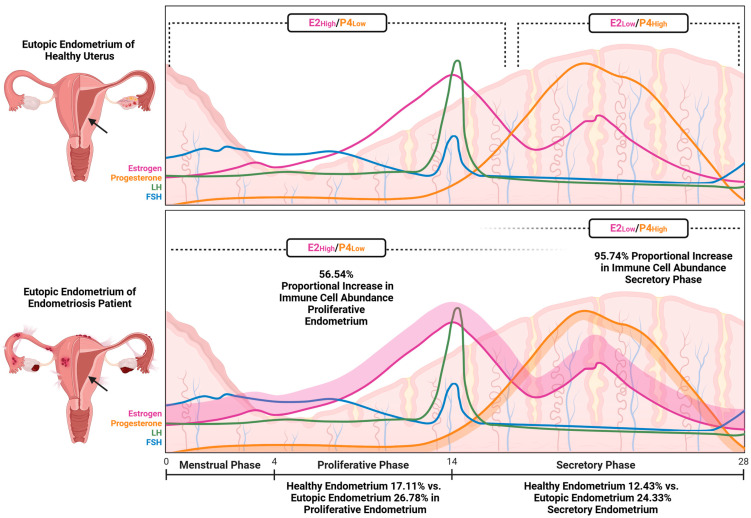
A summary of the hormonal changes in endometrial tissue, with reference to the proportional abundance of immune cells between the healthy endometrium versus eutopic endometrium in endometriosis. The menstrual cycle (0 to 28 days) is annotated to scale given the expected endometrial thickness, with corresponding estrogen (E2, pink), progesterone (P4, orange), luteinizing hormone (LH, green), and follicle-stimulating hormone (FSH, blue) levels overlaid. Wider lines indicate hormonal variance, while dashed lines to fade show uncertainty in hormonal crossovers in reference to the 28-day cycle. Proportional Increases were calculated from statistics derived from works reported in Huang, et al. [7].

**Figure 2 cells-14-00058-f002:**
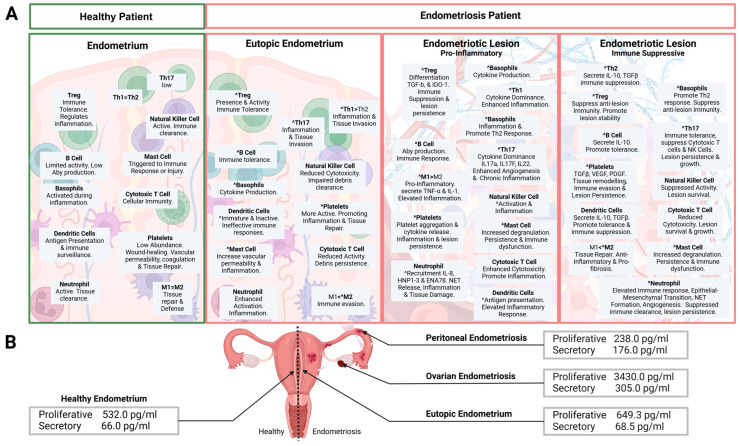
Immune cell dynamics and local estrogen concentrations in endometriosis. (**A**) Comparative immune cell profiles in the healthy endometrium, eutopic endometrium, and peritoneal lesions: pro-inflammatory vs. immune-suppressive microenvironments. The corresponding endometriosis patient panels, although not depicted directly, are under the pretext of a dysregulated endocrine milieu of estrogen dominance and progesterone resistance. The exact proportion and distribution of immune cell subtypes is still debated; however, there remain notable differences in activation states and function. ‘^’ indicates an elevation of relative cell abundance or activity. Key acronyms: macrophage phenotypes 1 or 2 (M1 and M2); T helper (Th); T regulatory (Treg). (**B**) Local E2 concentration in women with and without endometriosis. Local (intratissue) E2 concentrations derived from Huhtinen et al., 2012 [24]. Tissue-specific E2 levels illustrate the differences in focal concentrations between endometriotic lesions and control tissues. Notable differences in E2 concentrations across endometriotic subtypes are also evident. These differences in E2 concentrations highlight the importance of investigating the constitutive or induced expression profiles of specific estrogen receptors in focal immune cells.

**Figure 3 cells-14-00058-f003:**
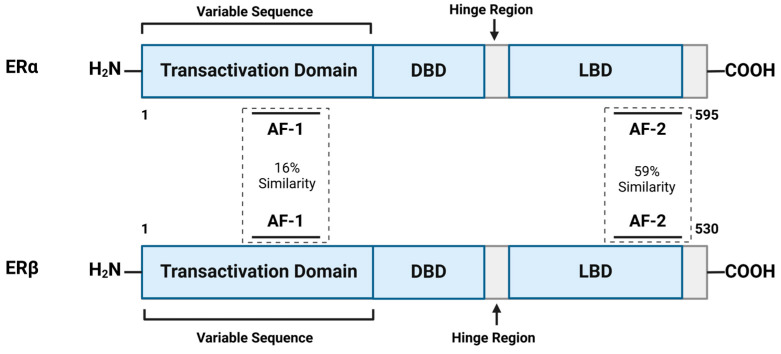
Structural summary of ER isoforms and their role in transcriptional regulation: differences between ERα and ERβ in AF1 and AF2 activity. The general structure of the ER includes the transactivation domain containing activation-function 1 (AF1), the DNA-binding domain (DBD), hinge region, ligand-binding domain (LBD), and activation-function 2 (AF2). ERs exist in two isoforms, ERα and ERβ, which form functionally distinct homo- and heterodimers (αα, αβ, and ββ) that play non-redundant roles in transcriptional regulation. The primary differences between ERα and ERβ are in the N-terminal domain, which contains the ligand-independent AF1, with a 16% similarity between the isoforms. The AF2 at the COOH-terminal ligand-binding domain (LBD) exhibits a 59% amino acid sequence similarity between the isoforms. Upon E2 binding, conformational change ensues to the receptor, enabling the recruitment of coregulators to the activation-function domains. The AF1 region, differing between ERα and ERβ, results in distinct transcriptional activities. In contrast, AF2 activity is similar in both isoforms.

**Figure 4 cells-14-00058-f004:**
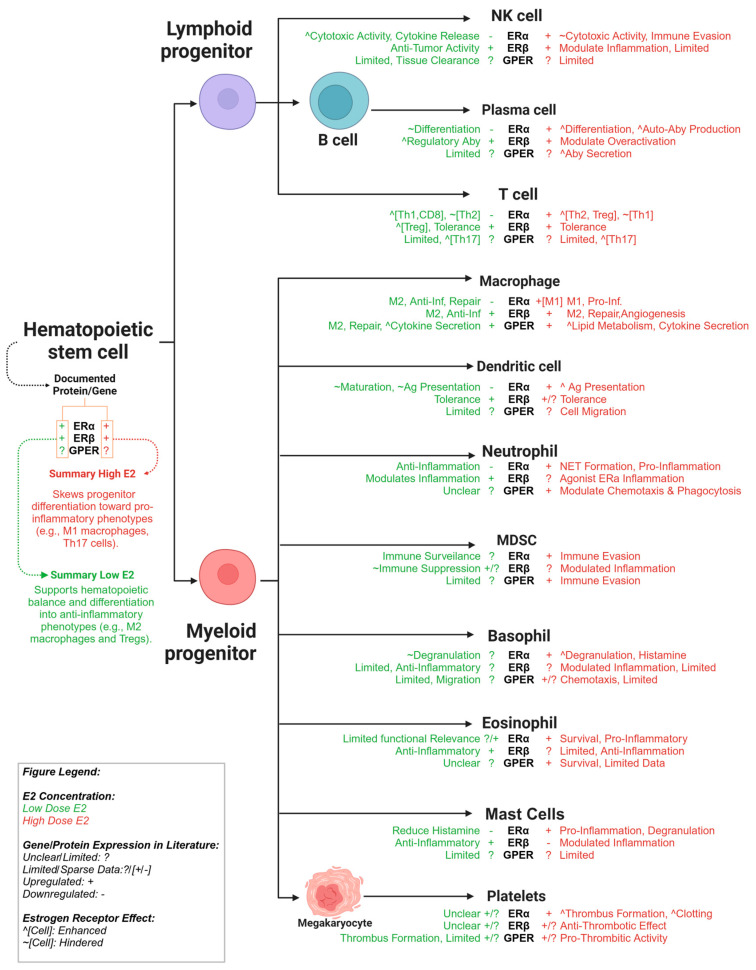
Effects of high and low estradiol (E2) on estrogen receptor expression and their respective roles in immune cells: A generalized snapshot of the current literature derived from endometrium and endometriosis tissue experiments. This figure provides a generalized overview of the effects of E2 on immune cells derived from endometrium and endometriosis tissues. The figure highlights how high and low E2 concentrations influence the expression of estrogen receptor alpha (Erα), beta (Erβ), and G protein-coupled estrogen receptor (GPER), as well as summarizes the resulting functions. Figure legend: E2 concentrations: high (red) and low (green); Gene/protein expression (respective to Erα/Erβ/GPER): unclear/limited information (‘?’); limited/sparse data with minimal findings (‘?’ alongside ‘+’ OR ‘−’); upregulated (+); downregulated (−). Note: up/downregulation predominantly refers to compared tissue regions inclusive of the internal (matched) endometrium and the healthy control endometrium. Estrogen receptor effect/function: enhanced (^‘Cell Name’); hindered (~‘Cell Name’).

## Data Availability

No new data were created or analyzed in this study.

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
