# Peer review of "The Estrogen–Immune Interface in Endometriosis"

_cells, 2025, doi:10.3390/cells14010058_

Round 1
Reviewer 1 Report
Comments and Suggestions for Authors
The working group has extensive experience in the study of endometriosis. The present review aims to reveal the interactions between estrogen and the immune system, highlighting their implications in the progression of endometriosis. The review is complete and is supported by a large number of investigations on the relationship between the immune system and estrogens, however, it is not possible to outline a clear molecular mechanism in the pathogenesis of endometriosis, due to the multiple activation pathways exerted by estrogens.
It is important to define the acronym RNA mentioned in line 39
Even though the introduction section lists the points addressed in this review, it is necessary for the authors to add a methodology section, where the parameters used for the search for articles are clearly described, such as words used, boolean operators, databases of the searches and the range of years of the articles used in said review.
It is recommended to use the abbreviation P4 for progesterone instead of PG, to avoid confusion with the abbreviation for prostaglandin (line 82)
In figure 1 it is not clear whether the hormone concentrations are the same in healthy patients vs those with endometriosis. It is also important to add the concentrations of FSH and LH as part of the changes in the menstrual cycle.
In section 4 (lines 181-219) a large number of cells of the immune system are mentioned; as well as hormonal receptors, whose interaction could be clearer if diagrams are made where the known regulatory pathways can be established and those where there are still questions, without a doubt that would allow the understanding of the mechanisms described in the normal and pathological regulation of uterine remodeling. I understand that in figure 2 the immune cells present in normal and pathological conditions are shown, so the next part would be to add the hormonal component to this image.
In figure 3, standardize the names of the alpha and beta estrogen receptors with Greek letters, as it appears in the figure caption.
In table 1, it would be more enriching to add the differences in the distribution of estrogen or progesterone receptors in the healthy endometrium vs those found so far in endometriosis and remove the columns of the genes that code for the receptors.
In line 471 the abbreviation iNOS should be defined and the definition of line 494 should be removed, leaving only the abbreviation there.
On line 582 separate the parenthesis from the word levels
On line 825 estrogens are defined as E2, however on line 81 it is already defined for the first time.
Author Response
Response to Reviewer #1 Comments
The working group has extensive experience in the study of endometriosis. The present review aims to reveal the interactions between estrogen and the immune system, highlighting their implications in the progression of endometriosis. The review is complete and is supported by a large number of investigations on the relationship between the immune system and estrogens, however, it is not possible to outline a clear molecular mechanism in the pathogenesis of endometriosis, due to the multiple activation pathways exerted by estrogens.
Comments 1. It is important to define the acronym RNA mentioned in line 39
Response 1. The acronym of RNA is now defined, and the revised sentence is “Issues such as cell isolation, low sensitivity to rare ribonucleic acid (RNA) populations embedded within extracellular matrices or tightly associated with neighboring cells, and non-uniform annotation continue to hinder our ability to assess transcript distribution and RNA processing roles”.
Comments 2. Even though the introduction section lists the points addressed in this review, it is necessary for the authors to add a methodology section, where the parameters used for the search for articles are clearly described, such as words used, boolean operators, databases of the searches and the range of years of the articles used in said review.
Response 2.
We thank the reviewer for this comment. We have added additional information on methodology in line 75-89.
“This literature review aims to enhance our understanding of hormonal-immune interactions in endometriosis by critically analyzing and summarizing existing research. We have used the following search terms in PubMed database: endometriosis AND immune cells; estrogen OR oestrogen AND Uterus; endometriosis NOT (estrogen OR oestrogen); (endometriosis OR estrogen OR oestrogen) AND immune cells; (estrogen OR oestrogen OR estrogen receptor OR oestrogen receptor) AND (endometriosis OR immune cells); endometriosis lesions AND immune cells; endometriosis AND immune cells NOT (estrogen OR oestrogen OR estrogen receptor OR oestrogen receptor); (estrog* OR oestro*) AND immune cells; ("immune cells" NEAR/5 estrogen) OR ("immune cells" NEAR/5 oestrogen). The searches also included additional terms (in the above same formatting): 'autoimmunity', 'monocytes', leukocytes', 'uterus', 'menstrual cycle', 'ovarian cycle', 'endometrial', 'female immunity'. Additional specific searches include for endometrium OR endometriosis AND specific immune cells (Neutrophil, Basophil, T cells etc.). This review present relevant theories, concepts, and findings to identify research gaps and provide a foundation for future studies.”
It must be noted that our review is not a systematic review or a meta-analysis. Unlike a systematic review, we did not attempt to answer a focused research question by synthesizing only specific data. Instead, we provide a summary of the existing literature on hormonal-immune interactions in endometriosis by examining all relevant available evidence. Similarly, unlike a meta-analysis, we did not aim to estimate the effect size of interventions or exposures.
Comments 3. It is recommended to use the abbreviation P4 for progesterone instead of PG, to avoid confusion with the abbreviation for prostaglandin (line 82)
Response 3. We have changed the abbreviation for progesterone from PG to P4 to avoid confusion with the abbreviation for prostaglandin, as kindly suggested by the reviewer.
Comments 4. In figure 1 it is not clear whether the hormone concentrations are the same in healthy patients vs those with endometriosis. It is also important to add the concentrations of FSH and LH as part of the changes in the menstrual cycle.
Response 4. Thank you for this comment. We have included the requested changes, summarising the hormonal changes in endometrial tissue, with reference to proportional abundance of immune cells between Healthy Endometrium versus Eutopic Endometrium in Endometriosis. We have also included additional information on FSH and LH. See updated Figure 1.
Comments 5. In section 4 (lines 181-219) a large number of cells of the immune system are mentioned; as well as hormonal receptors, whose interaction could be clearer if diagrams are made where the known regulatory pathways can be established and those where there are still questions, without a doubt that would allow the understanding of the mechanisms described in the normal and pathological regulation of uterine remodeling. I understand that in figure 2 the immune cells present in normal and pathological conditions are shown, so the next part would be to add the hormonal component to this image.
Response 5. Thank you for this comment. We have included an additional, Figure 4, as requested. This new figure summarises the effects of high and low estradiol (E2) on estrogen receptor expression and their respective roles in immune cells, derived from literature examining endometrium and endometriosis tissue.
Comments 6. In figure 3, standardize the names of the alpha and beta estrogen receptors with Greek letters, as it appears in the figure caption.
Response 6. As per feedback, we have amended the figure to be consistent with the figure citations to include the alpha and beta symbols of the Greek alphabet.
Comments 7. In table 1, it would be more enriching to add the differences in the distribution of estrogen or progesterone receptors in the healthy endometrium vs those found so far in endometriosis and remove the columns of the genes that code for the receptors.
Response 7. As per advice for Table 1, the columns for the genes that encode the oestrogen receptors have been removed. We agree with the reviewer that it will be more enriching to add the differences in oestrogen receptors expression comparing healthy endometrium vs endometriosis; however, to the best of our knowledge, there are limited reports that directly compared estrogen receptor expression in the specific immune cells comparing healthy endometrium vs endometriosis. Such studies will require identification of the immune cells in healthy endometrium and endometriosis; and an additional layer to determine the levels of ESR1, ESR2 or GPER1. In fact, single-cell transcriptomics studies may be useful for this purpose, though there is still limited analysis to date.
Comments 8. In line 471 the abbreviation iNOS should be defined and the definition of line 494 should be removed, leaving only the abbreviation there.
Response 8. Revised as kindly suggested by the reviewer.
Comments 9. On line 582 separate the parenthesis from the word levels
Response 9. Revised as kindly suggested by the reviewer.
Comments 10. On line 825 estrogens are defined as E2, however on line 81 it is already defined for the first time.
Response 10. Revised as kindly suggested by the reviewer.
Reviewer 2 Report
Comments and Suggestions for Authors
Dear authors,
Thank you for the opportunity to review the study "The Estrogen-Immune Interface in Endometriosis" by Greygoose et al. The study, which focuses on the interaction between estrogen and the immune system in the context of endometriosis, explores how these interactions contribute to the progression of the disease and their impact on the immune system in healthy and affected tissues.
This is a topic of great relevance, which fills a crucial gap in knowledge about the hormonal and immunological interaction in endometriosis, offering new perspectives on little explored mechanisms that influence the progression of the disease.
Below are some suggestions:
1- I suggest that the authors include a more in-depth discussion on how data from multiomics studies (such as transcriptomics and proteomics) could be used to identify new biomarkers or pathological mechanisms.
2- It would be interesting to add a section on the clinical implications of these findings, exploring potential therapeutic targets or diagnostic tools.
3- I ​​suggest the inclusion of a "Limitations" section, where the limitations of the study can be discussed, such as the possibility of extrapolating the results of in vitro studies to clinical practice, the relationship between hormonal levels and clinical effects, and the connection between the findings and their applicability in clinical practice.
4- I believe it would be relevant to highlight the challenges of current research and suggest directions for future studies, such as the need to investigate the heterogeneity of immunological responses in the disease.
5- Regarding the figures, they could be improved with more detailed captions about the immunological markers and their relevance in the contexts presented.
Thank you again for the opportunity to review this manuscript.
Author Response
Response to Reviewer #2 Comments
Dear authors,
Thank you for the opportunity to review the study "The Estrogen-Immune Interface in Endometriosis" by Greygoose et al. The study, which focuses on the interaction between estrogen and the immune system in the context of endometriosis, explores how these interactions contribute to the progression of the disease and their impact on the immune system in healthy and affected tissues.
This is a topic of great relevance, which fills a crucial gap in knowledge about the hormonal and immunological interaction in endometriosis, offering new perspectives on little explored mechanisms that influence the progression of the disease.
Below are some suggestions:
Comments 1. I suggest that the authors include a more in-depth discussion on how data from multiomics studies (such as transcriptomics and proteomics) could be used to identify new biomarkers or pathological mechanisms.
Comments 2. It would be interesting to add a section on the clinical implications of these findings, exploring potential therapeutic targets or diagnostic tools.
Response 1 and 2- The authors would like to thank the reviewer for the positive feedback, which has allowed us to improve our manuscript.
We incorporated the following paragraph in introduction, per request of the Reviewer, starting from line 43. “Integrating multiomics data, particularly combining transcriptomics and proteomics, may offer significant promise for identifying novel biomarkers and elucidating the pathological mechanisms of endometriosis. By analyzing mRNA expression (from transcriptomics) and protein abundance (from proteomics) in endometriosis tissues, differentially expressed genes and proteins can be identified, advancing diagnostic markers and insights into disease progression. This approach also enables pathway analysis, revealing disrupted molecular pathways and protein-protein interaction networks, which are crucial for understanding the disease pathophysiology and identifying potential therapeutic targets. Moreover, multiomics can help address disease heterogeneity by uncovering distinct RNA and protein expression patterns across endometriosis subtypes. However, several challenges must be addressed to fully realize the potential of multiomics in endometriosis research. Data complexity, tissue availability, and the inherent heterogeneity of the disease complicate the interpretation of results. In particular, the integration of diverse omics data requires robust technical standardization. Developing standardized protocols for sample collection, processing, and data analysis is essential to ensure reproducibility and enable cross-study comparisons. These efforts are critical for advancing both the diagnostic and therapeutic potential of multiomics in endometriosis”
Comments 3. I ​​suggest the inclusion of a "Limitations" section, where the limitations of the study can be discussed, such as the possibility of extrapolating the results of in vitro studies to clinical practice, the relationship between hormonal levels and clinical effects, and the connection between the findings and their applicability in clinical practice.
Comments 4. I believe it would be relevant to highlight the challenges of current research and suggest directions for future studies, such as the need to investigate the heterogeneity of immunological responses in the disease.
Response 3 and 4. We have added limitation part to the conclusion paragraph and revised as conclusions and limitations, and incorportated the following paragraph as limitations (line 921): “Despite these insights, there are several limitations that need to be considered. First, extrapolating findings from in vitro studies to clinical practice remains a challenge, as laboratory conditions may not fully replicate the complexity of in vivo environments. Additionally, while hormonal imbalances are well-documented in endometriosis, the precise relationship between hormonal levels and their clinical effects—such as symptom severity and disease progression—remains unclear. Not only the heterogeneity of endometriosis itself, but also the heterogeneity of immune responses in endometriosis further complicates the interpretation of results, as different subtypes of the disease may exhibit distinct immune signatures even in different phases of the disease. Moreover, the availability and accessibility of patient tissue samples limit the scope of some studies.
Finally, the applicability of current findings in clinical practice is limited by the lack of standardized protocols for sample collection, processing, and analysis, which hampers reproducibility and comparison across studies. Moving forward, future research should aim to clarify the immunological heterogeneity of the disease, focusing on both the functional roles of immune cell subsets and the influence of hormones on their behavior. Additionally, studies should seek to integrate clinical data with molecular findings to better understand the relationship between immune dysregulation and disease symptoms. These efforts will be essential for developing more effective and personalized therapeutic strategies for endometriosis.”
Comments 5. Regarding the figures, they could be improved with more detailed captions about the immunological markers and their relevance in the contexts presented.
Response 5. We have amended the captions to include more information.
Reviewer 3 Report
Comments and Suggestions for Authors
The authors have described the relationship between estrogens, the immune system and endometriosis. I believe the manuscript is well-written and very precise regarding the in-depth concepts. The references are adequate, as is the depth of the content.
However, some information regarding the role of GPER in endometriosis should be added in paragraph 6, as it is mentioned but not described within the pathology. Literature regarding the GPER receptor in the context of endometriosis should be added.
Adding a graphical abstract could be useful for the understanding of the article.
I believe the manuscript is an exhaustive summary of the relationship between the immune system and estrogen in the context of endometriosis.
Author Response
Response to Reviewer #3 Comments
Comments 1. The authors have described the relationship between estrogens, the immune system and endometriosis. I believe the manuscript is well-written and very precise regarding the in-depth concepts. The references are adequate, as is the depth of the content.
However, some information regarding the role of GPER in endometriosis should be added in paragraph 6, as it is mentioned but not described within the pathology. Literature regarding the GPER receptor in the context of endometriosis should be added.
Response 1. We thank the reviewer for the positive comment. We have added information regarding the role of GPER in endometriosis in line 440 as follow:
“GPER1 has a unique pattern of expression in endometriosis. A study consisted of 74 ovarian, peritoneal and deep infiltrating endometriosis showed high cytoplasmic GPER1 expression levels in the epithelial component of endometriosis and none in the normal endometrium. There was no significant difference in nuclear epithelial GPER1. Endometrioma has the highest frequency of cytoplasmic epithelial GPER1 as compared to peritoneal or deep infiltrating endometriotic lesions. GPER overexpression in en-dometriosis suggests roles in hormonal regulation. The inhibition of GPER in ectopic endometrial stromal cells using miRNAs reduced proliferation, migration and invasion [24, 52, 53]. These data suggest the importance of GPER in endometriosis progression.”
Comments 2. Adding a graphical abstract could be useful for the understanding of the article.
Response 2. Thank you for this comment. We have now added a graphical abstract.
Comments 3. I believe the manuscript is an exhaustive summary of the relationship between the immune system and estrogen in the context of endometriosis.
Response 3. The authors would like to thank the reviewer for their positive feedback, which has allowed us to improve our manuscript.